# Study of *Camellia sinensis* diploid and triploid leaf development mechanism based on transcriptome and leaf characteristics

**Xinzhuan Yao**[1], **Yong Qi**[2], **Hufang Chen**[1], **Baohui Zhang**[1], **Zhengwu Chen**[2], **Litang Lu**[1,3]*

**1** College of Tea Science, Guizhou University, Guiyang, Guizhou, People's Republic of China, **2** Guizhou Academy of Agricultural Sciences, Guiyang, China, **3** The Key Laboratory of Plant Resources Conservation and Germplasm Innovation in Mountainous Region (Ministry of Education), Institute of Agro-Bioengineering, Guiyang, Guizhou, People's Republic of China

* ltlv@gzu.edu.cn

**Data Availability Statement:** Data are available from the National Genome Science Data Center (https://ngdc.cncb.ac.cn/), Project No: PRJCA011051 Submission No. SubPRO016312.

## Abstract

Polyploidization results in significant changes in the morphology and physiology of plants, with increased growth rate and genetic gains as the number of chromosomes increases. In this study, the leaf functional traits, photosynthetic characteristics, leaf cell structure and transcriptome of *Camellia sinensis* were analyzed. The results showed that triploid tea had a significant growth advantage over diploid tea, the leaf area was 59.81% larger, and the photosynthetic capacity was greater. The morphological structure of triploid leaves was significantly different, the xylem of the veins was more developed, the cell gap between the palisade tissue and the sponge tissue was larger and the stomata of the triploid leaves were also larger. Transcriptome sequencing analysis revealed that in triploid tea, the changes in leaf morphology and physiological characteristics were affected by the expression of certain key regulatory genes. We identified a large number of genes that may play important roles in leaf development, especially genes involved in photosynthesis, cell division, hormone synthesis and stomata development. This research will enhance our understanding of the molecular mechanism underlying tea and stomata development and provide a basis for molecular breeding of high-quality and high-yield tea varieties.

## Introduction

Polyploid plants are new species that develop with more than two complete sets of chromosomes in their genomes, which can occur through artificial synthesis or natural domestication. Generally, as the ploidy of the chromosomes increases, the volume of the plant cells and nuclei becomes correspondingly larger, and the leaf growth morphology index, pollen, fruit, seed size and stem diameter increase accordingly. The characteristics of polyploid plants are known as Kyodaika [1,2]. In the epidermal cells of *Arabidopsis thaliana*, the size of the cells and the DNA content in the nucleus are proportional. The 4C DNA content is multiplied, and the corresponding *Arabidopsis thaliana* cell volume is larger [2]. In terms of biomass, polyploid plants are generally taller, grow faster and have more biomass than diploid plants [3].

**Funding:** This work was supported by the Technology Creation Center of Guizhou Tea Industrialization (Qiankezhongyindi [2017]4005); Guizhou Science and Technology Plan Project (Qiankehe Support 2021 General 111).

**Competing interests:** The authors have declared that no competing interests exist.

Polyploidization has certain effects on plant physiology and development, including the water balance, photosynthetic rate, carbon dioxide exchange rate and hormone levels [4,5]. In theory, an increase in the number of plant chromosomes should result in longer periods of cell division and slower physiological processes in many plants [6]. However, previous studies showed that hexaploid wheat has stronger metabolism and faster growth than tetraploid wheat, with more rapid root growth, mitosis and biomass storage as well as greater photosynthesis capacity [7]. The doubling of chromosomes first causes a series of changes in the plant genome, which in turn affects the growth and development of the plant [8]. The alteration of the whole genome in polyploid plants leads to changes in the genomic structure, resulting in the re-regulation of gene expression and changes in gene expression levels. A large amount of research has shown that plant polyploidization causes changes in genome interactions, DNA sequences and gene expression [9]. Kashkush used complementary DNA-amyloid precursor-like protein (cDNA-APLP) technology to determine the expression levels of allopolyploid and natural polyploid genes in *Arabidopsis*, cotton and wheat [10]. However, in tea plants, which are mainly grown for their vegetative organs, polyploidy is more advantageous than diploidy. In addition to the larger organs, the photosynthesis and stress resistance of polyploid plants are often greater than those of diploids. For example, polyploid plants of wheat [11], soybean [12] and other plants have stronger photosynthesis and resistance than the corresponding diploid plants.

*Camellia sinensis* is an evergreen economic crop with perennial leaves. It is widely distributed in Guizhou, Yunnan, Sichuan and other provinces in China. Guizhou has the largest tea tree planting area in the country. The characteristics of tea leaves (leaf area, leaf shape index, photosynthetic pigment content, etc.) are closely related to the growth and development of the tree, and photosynthesis is the basis for the accumulation of dry matter and the chemical quality of tea leaves. Previous studies have shown that there is a large biological diversity in chemical quality, leaf morphology and physiological activity among plants in different genetic groups (wild or cultivated species) [13,14]. However, these studies did not thoroughly explore the internal connection between plant functional traits and photosynthetic physiology. In this study, QianMei 419 and QianFu 4 [15] were used as research materials to explore the mechanism of leaf development based on transcriptome technology and leaf characteristics, with a view toward breeding new tea varieties, understanding the utilization of tea leaf resources and collecting scientific data.

## Materials and methods

### Plant materials and growth conditions

The diploid tea tree 'QianMei 419' is a small and medium-sized leaf breed cultivated by the Tea Science Institute of Guizhou Academy of Agricultural Sciences. The triploid 'QianFu No. 4' was obtained by using the Co60-γ ray to mutate the seeds of 'QianMei 419' strictly self-pollinated by the Institute of Tea, Guizhou Academy of Agricultural Sciences. Chen et al. [15] studied mitosis in the leaves of QianFu No. 4 and found 45 chromosomes, which is one more group than the number in 'QianMei 419'. For this study, 'QianMei 419' and 'QianFu No. 4' were planted in tea gardens through cuttings in 2015 (Guizhou Institute of Agricultural Sciences, Guizhou Academy of Agricultural Sciences Tea Institute—Tea Lake Institute, north latitude 27.76, east longitude 107.49, altitude 760 m above mean sea level).

We selected triploid and diploid materials that grew rapidly and uniformly with no pathogen infection. One bud and three leaves were selected from the tea trees. Some of the materials were quick-frozen in liquid nitrogen and stored in a refrigerator at -80˚C for later use, and the rest were used for cytology studies.

## Observation and determination of leaf epidermal cells

Scissors were used to cut off complete and strong leaves from diploid and triploid tea trees. After washing with water, the leaf epidermis was removed. First, a blade was used to lightly draw a small square on the leaf surface, and the epidermis was separated with tweezers. The upper epidermis of the leaf was removed, placed on a glass slide dripped with distilled water and flattened with no overlaps. The slide was covered with a cover glass and placed under a microscope for observation. Pictures were obtained with the Leica DFC450 microscope. The number of stomata in the field of view was recorded, and 30 stomata were randomly selected. The Leica LAS software was used to measure the size of the stomata and calculate the stomata density.

## Observation and measurement of leaf anatomical structure

Using the conventional paraffin sectioning method, tea tree diploid and triploid leaves were fixed with formalin acetic acid alcohol (FAA), dehydrated with gradient ethanol, and embedded in paraffin. The leaves were cut into 8–10 μm-thick sections with a Leica RM2235 microtome, counterstained with safranine and fast green dye, and observed under a Leica DM 4000B microscope. Pictures were obtained with a Leica DFC450 microscope. Leica LAS software was used to measure the thickness of the tea leaves and the thickness of each tissue layer of the tea tree as well as for data analysis.

## Functional annotation, classification and metabolic pathway analysis

All assembled unigenes were annotated using BLASTx alignment (E-value $\leq 10^{-5}$) based on the following databases: National Centre for Biotechnology Information (NCBI) Non-Redundant (NR) database and Nucleotide Collection (Nr/Nt), Swiss-Prot, Gene Ontology (GO), Kyoto Encyclopedia of Genes and Genomes (KEGG) and the Clusters of Orthologous Groups of proteins (COG). Using the NR annotation results, GO function annotation of the unigenes was performed with Blast2GO (v2.5.0). After obtaining the GO annotation for each unigene, the WEGO software was used to perform GO function classification for all unigenes, and the gene function distribution characteristics of the species were macroscopically determined. GO and KEGG pathway enrichment analyses for the differentially expressed unigenes were then conducted. The obtained GO annotations were enriched and refined using the topGo package (v2.16.0). The biologically complex behaviors of the genes were further studied using KEGG, and the pathway annotations for the unigenes were obtained from the KEGG annotation information. The read counts were normalized by calculating the number of reads per kilobase per million (RPKM) for each transcript in the individual tissues, and the relative expression of the genes was calculated using $Log_2$ (YH29/WH10). A P-value cut-off of $\leq 0.05$ along with at least two-fold change was used to identify significant differential expression of the transcripts.

## Differential expression and enrichment analysis

The abundance of all genes was normalized and calculated using uniquely mapped reads with the RPKM method [16]. Read count was normalized with the trimmed mean of M-values method, and differential expression analysis was performed using the DEGseq package with a threshold p-value of $< 0.05$ and fold change of $> 2$ demarcating significantly different expression levels. GO and KO enrichment analyses were performed based on the identified differentially expressed genes.

### Gene validation and expression analysis

For the purpose of gene validation and expression analysis, all the DEGs related to leaf growth were subjected to quantitative real-time polymerase chain reaction (qRT-PCR). qRT-PCR was performed using a 7500 Fast Quantitative Real-Time PCR instrument (Applied Biosystems, Waltham, Massachusetts, USA) and SYBR Premix Ex Taq (TaKaRa) according to the manufacturer's instructions. The synthesized cDNA was used for the analysis of transcript abundance using qRT-PCR and the primers shown in S1 Table. The relative levels of the transcripts were determined by normalizing the expression against actin transcript levels. Experiments were replicated three times.

### Statistical analysis

All data were expressed as mean ± standard error. For qRT-PCR, three biological replicates were assessed. Microsoft Excel and GraphPad Prism 5.0 software were used for data analysis. One-way ANOVA with Duncan's multiple range test was used for post hoc comparison of multiple variables. A significant difference relative to the control was recognized at $^{*}P < 0.05$ or $^{**}P < 0.01$.

## Results

### Analysis of the number and size of stomata in diploid and triploid leaves

In order to investigate the differences in stomatal appearance on the epidermis of diploid and triploid tea leaves, we performed statistical analyses of the size and density of the stomata of the epidermis in the same parts of the tea leaves. The results showed that the stomata size (length and width) of triploid leaves was significantly larger than that of diploid leaves, with an increase of 57.20% and 84.44%, respectively, compared to the diploid length and width. However, the stomata density of triploid leaves was significantly lower than that of diploids; the stomata density of diploid tea leaves was twice that of triploid stomata (**Fig 1**).

### Anatomical structure analysis of diploid and triploid leaves

We evaluated paraffin sections of tree leaves to understand the differences in the growth and development of diploid and triploid leaves. Fig 2 shows that the leaf veins of diploid tea plants underwent significant changes after triploidization, with changes in the xylem of the veins being the most obvious. The xylem of the triploid leaf vein was more developed and larger than that of the diploid. The area of triploid xylem was three times that of the diploid at 0.476 mm$^2$. There are two reasons for the increase in the area of the triploid xylem: one is the increase in the cell area; the other is the increase in the number of xylem cells. The average number of cell layers in the diploid and triploid xylem was 19 and 25, respectively. The number of xylem cell layers in triploids was 30.67% greater than that of diploids. There were no significant differences in the size of the diploid and triploid veins, as well as the phloem and formation.

It can be seen from Fig 3 that the shape and size of diploid and triploid mesophyll cells were significantly different. The thickness of epidermal cells in triploid mesophyll was 22.28% larger than that of the diploid, which is extremely significant. It can be seen from the figure that the fence organization in diploid leaves was tighter, and the shape and size were relatively uniform. The palisade tissue cells of the triploid tea leaves were large, the arrangement was relatively tight, and the sponge cell gap was large (the section may have been partially broken, which would have made it difficult to observe cell breakage on the cross-section of the leaf). The

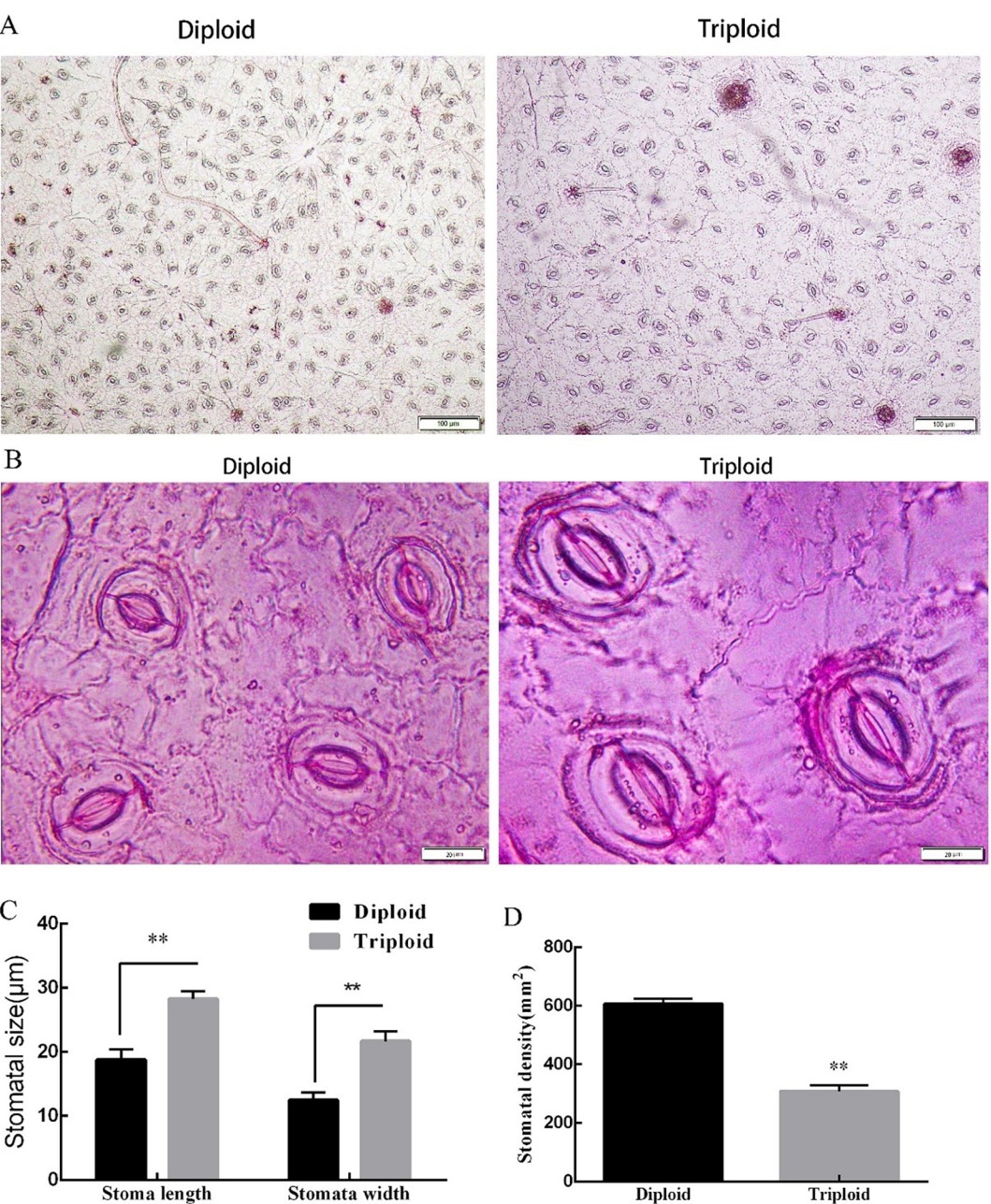

**Fig 1. Number of stomata in diploid and triploid tea leaves.** A: Number of diploid stomata and triploid stomata at 100X magnification; B: Number of diploid stomata and triploid stomata at 1000X magnification; C: Length and width of diploid and triploid stomata; D: Diploid and triploid stomatal density in the same field of view. Error bars indicate SD (n = 3); statistical significance is indicated $^*P<0.05$, $^{**}P<0.01$.

average length of the diploid palisade tissue cells was greater than that of the triploid cells, with a length of 65 μm, which was 15.65% longer than that of the triploid cells. The width of triploid palisade tissue cells was significantly larger by 70% compared to that of diploid cells. Diploid sponge tissue cells were small, dense and about twice the number of triploid cells, while triploid sponge tissue cells were large and had large intercellular spaces.

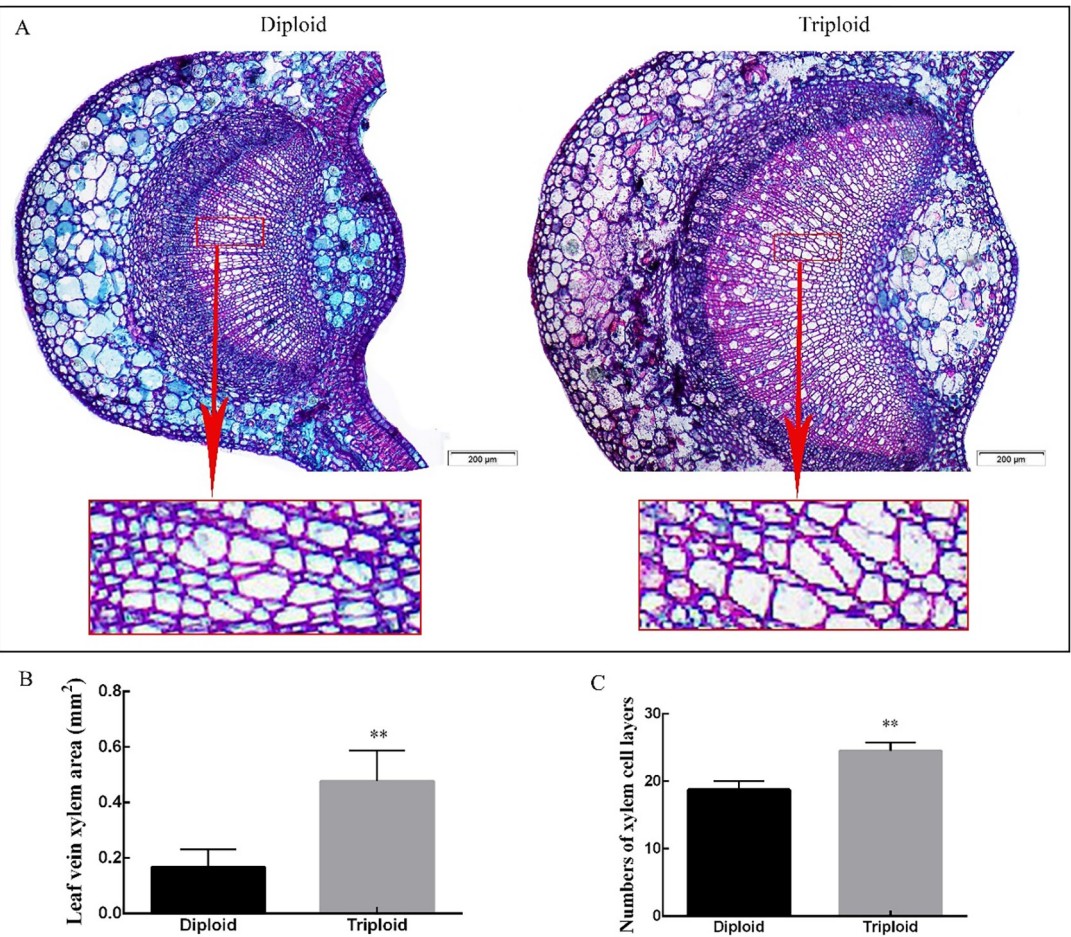

**Fig 2. Transverse section of tea diploid and triploid leaf veins.** A: Diploid and triploid vein xylem cross section under 50X magnification. B&C: Statistical data for cross sections of diploid and triploid leaf vein xylem blades. Error bars indicate SD (n = 3); ** indicates significant difference at the 0.01 level between the diploid and triploid.

## Illumina sequencing and reads assembly

To investigate the molecular mechanisms of diploid and triploid leaf growth and to understand the metabolic processes involved in leaf growth and development, we analyzed the gene expression profiles of diploid and triploid leaves. Using de novo transcriptome sequencing, an average of about 40 million original reads was obtained for the test samples, with high quality reads reaching over 99%. Six cDNA libraries (CaS419_1, CaS419_2, CaS419_3, CaS4_1, CaS4_2, and CaS4_3) were generated from diploid and triploid mRNAs and sequenced using the Illumina deep-sequencing HiSeq ™ 2000.

The raw data obtained after sequencing on the machine were filtered, and the sequencing error rate was assessed. The guanine-cytosine (GC) content distribution was also evaluated, and GC content analysis was used to identify A/T or G/C separation phenomenon. Finally, data for six samples (clean reads) were obtained for subsequent analysis. The filtered data are summarized in the Table 4. Ninety-six percent of the raw reads had Phred-like quality scores at the Q20 level (an error probability of 1%). After removing adapters, low-quality sequences and ambiguous reads, we obtained approximately 45 million, 43 million and 60 million clean reads from the diploid samples (CaS419_1, CaS419_2 and CaS419_3) and 58 million, 62 million and 48 million from the triploid samples (CaS4_1, CaS4_2 and CaS4_3), respectively. Raw reads were filtered

and assembled using the de novo assembly software Trinity. The assembled sequences were found to be redundant and were spliced using the TGICL software to obtain the longest non-redundant unigene set. Further statistical analysis was performed on the unigene sets.

## Functional annotation and cluster analysis

Only 27,031 unigenes were co-annotated into six databases (NR, NT, SwissProt, COG, GO, and KEGG), accounting for 26.13% of 103,448 unigenes. Among them, 90,547 and 89,933 unigenes (87.53% and 86.93% of all annotated unigenes) were most frequently cited in the NCBI NR and NT databases, and 35,298 (34.12%) and 61,318 (59.27%) unigenes could be annotated into the COG and Swiss-Prot databases. We annotated 45,820 (44.29%) and 67,980 (65.71%) unigenes into the GO and KEGG databases (Fig 4).

The main GO terms were biological process (BP), cellular component (CC), and molecular function (MF). Based on sequence homology, 45,820 unigenes were mainly categorized into

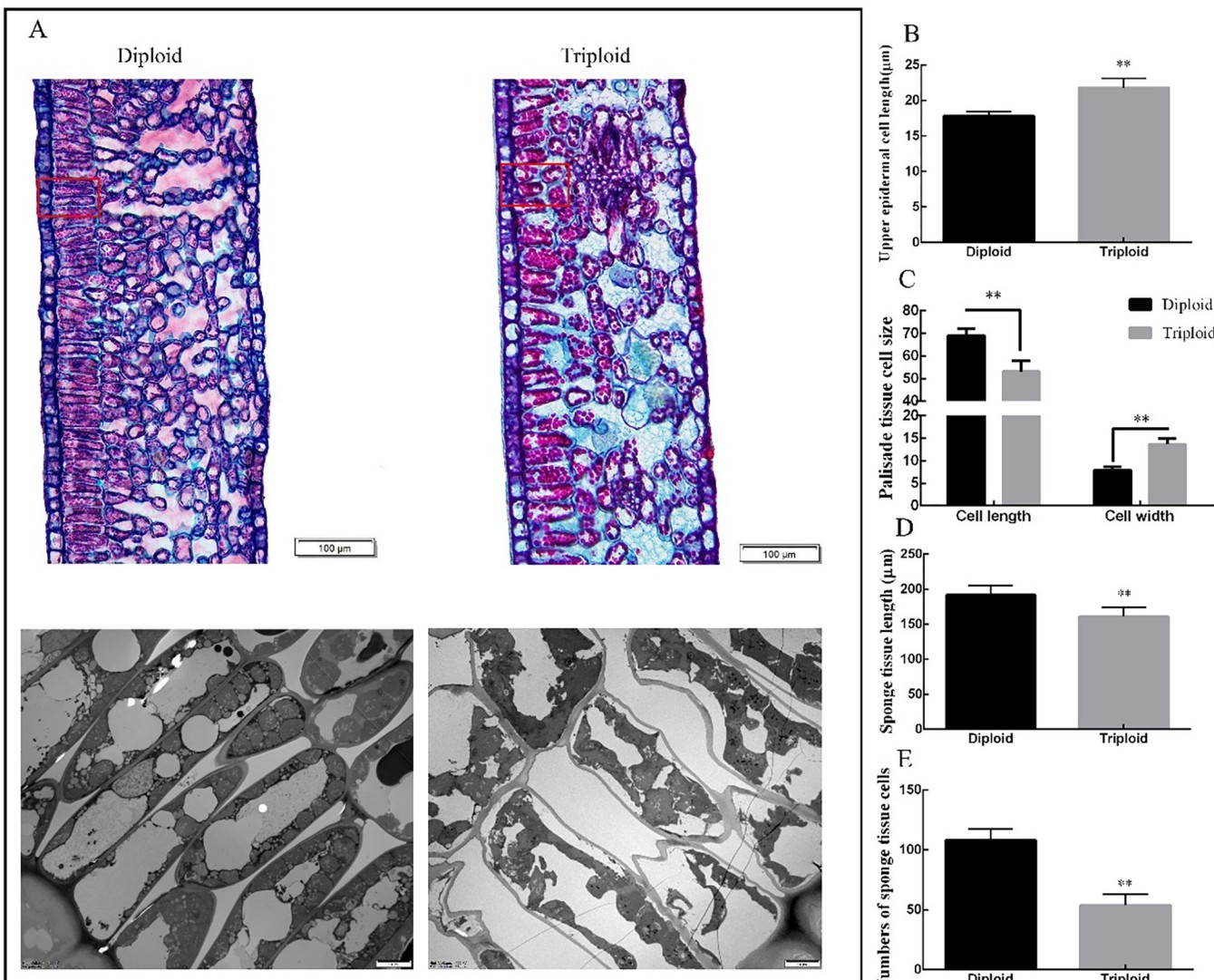

**Fig 3. Transverse section of tea diploid and triploid leaf mesophyll.** A: Diploid and triploid mesophyll cross sections under 100X magnification. B-E: Cross sectional length of upper epidermal cell, size of the palisade tissue cell, number of sponge tissue cells and sponge tissue length. Error bars indicate SD (n = 3); ** indicates significant difference at the 0.01 level between the diploid and triploid.

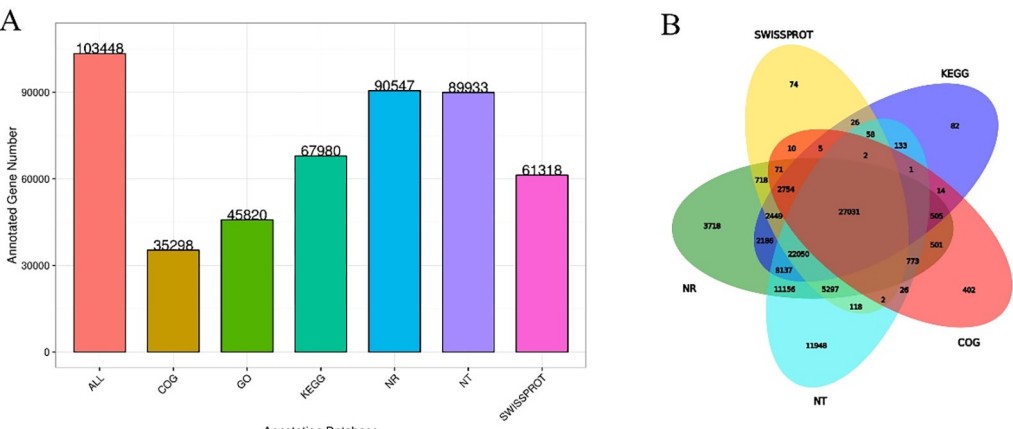

**Fig 4. Unigene database statistics, functional annotations and Venn diagram.** A: The X-axis represents the database, where All represents the union of all the data; the Y-axis represents the number of unigenes in the corresponding database annotation, and the number in the figure corresponds to the number of unigenes in the annotation. B: Venn diagram showing the number of specific genes between the two samples.

55 functional groups (**Fig 5**). In the BP category, the two major groups, cellular processes and metabolic processes, accounted for the highest proportion. Of these, approximately 24,750 genes were annotated under the metabolic processes category, which may allow for the identification of novel genes involved in secondary metabolism pathways in triploid trees. In the MF category, unigenes with binding and catalytic activity formed the largest groups. For CC, the top three largest categories were cell, cell parts, and membranes. To further evaluate the reliability of our transcriptome results and the effectiveness of our annotation process, we searched the annotated sequences for genes with COG classifications (**Fig 6**). Among the 26 COG categories, the cluster for "General function prediction only" (9,286) represented the largest group, followed by "Transcription" (5,108), "Posttranslational modification, protein

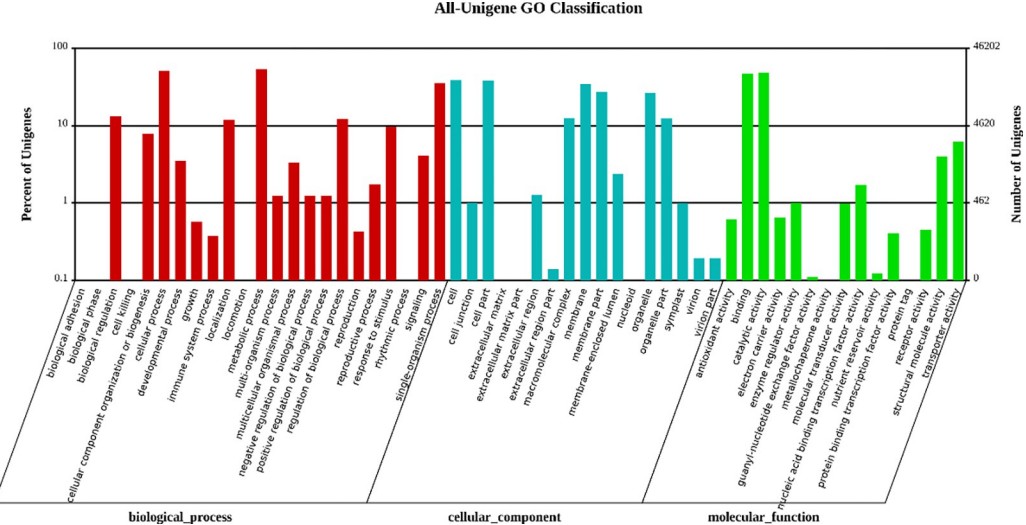

**Fig 5. GO classification map.** The abscissa is the GO term for the next level of the three major categories in GO, and the ordinate is the number of genes annotated to a term (including the subterm of the term). The three different categories represent the three basic tree classifications for GO terms (from left to right, biological processes, cellular components, molecular functions).

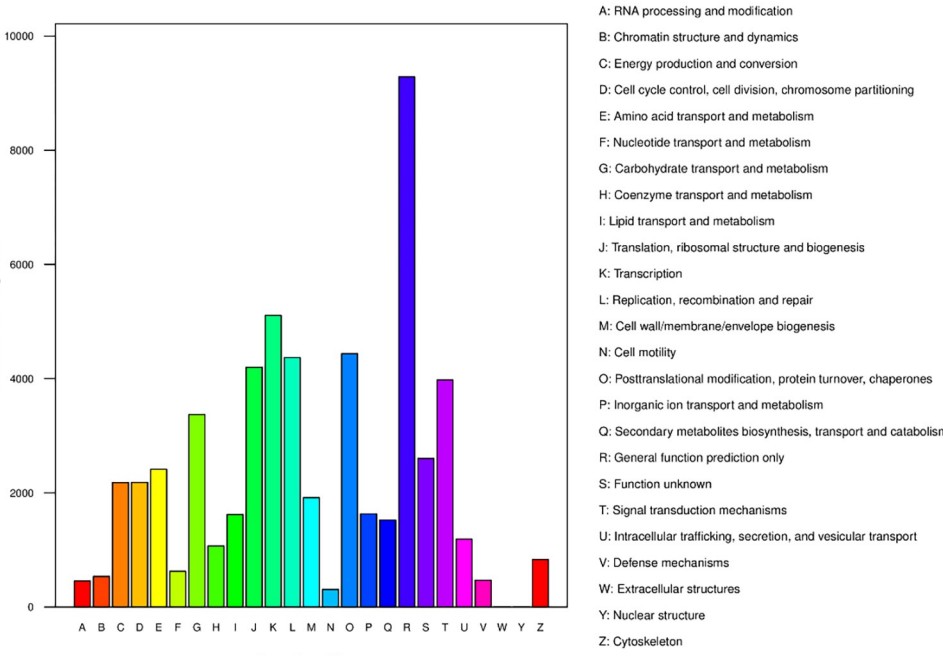

**Fig 6. COG function classifications of unigenes for Clari as macrocephalus.** Left-axis indicates the number of unigenes; letters on the x-axis represent different COG categories.

turnover, chaperones" (4,438), and "Replication, recombination and repair" (4,369). The category "Extracellular structures" (6) was the smallest group.

To determine the biological functions of the unigenes, we compared the annotated sequences against the KEGG database. In total, 67,980 annotated unigenes were assigned to 136 known pathways based on the KEGG BLAST analysis. The majority of the unigenes (22,658; 31.95%) were involved in the global and overview maps pathways, followed by Carbohydrate metabolism (7,072 unigenes; 9.97%), Translation (6,418 unigenes; 9.05%), and Folding, sorting and degradation (4,408 unigenes; 6.22%).

## Differentially expressed genes and qRT-PCR validation between diploid and triploid *Camellia sinensis*

To confirm the results of the Solexa/Illumina sequencing, 12 unigenes were selected for quantitative RT-PCR assays. The qRT-PCR analysis performed for ten upregulated and two down-regulated growth-related DEGs confirmed the transcriptomic changes detected with RNA-seq (**Fig 7**). Based on the biological replications, a total of 103,348 unigenes were generated and 23,813 DEGs between QianMei 419 and QianFu 4 leaves were identified. Among the DEGs, 16,459 were up-regulated and 7,354 DEGs were down-regulated (S2 Table). qRT-PCR indicated that all of the selected unigenes exhibited expression patterns similar to those obtained from the transcriptome sequencing analysis, suggesting that the transcriptome profiles accurately reflected the global transcriptome differences between QianMei 419 and QianFu 4.

## Major pathways involved in leaf development

DEG KEGG pathway analysis only shows the top 20 pathways enriched to the most pathway entries (**Fig 8** and S3 Table). Of the 20 pathways, only five pathways (Photosynthesis, Plant-

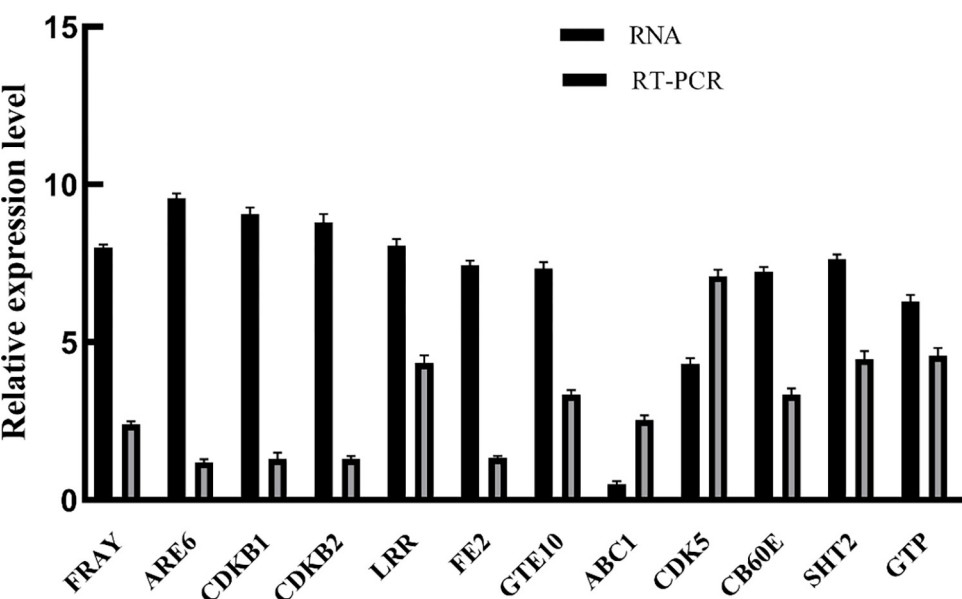

**Fig 7. Expression patterns of genes related to growth and development of diploid and triploid tea leaves analyzed by qRT-PCR.**

pathogen interactions, Photosynthesis—Antenna Proteins, and Oxidative phosphorylation signaling system) were related to plants. Among these pathways, photosynthesis is the most important pathway in plants. From the 74 high-quality transcripts identified from the photosynthetic pathway (68 up-regulated, 6 down-regulated), the transcriptome encodes 40 protein families (Table 1). The photosynthetic pathway is mainly divided into photosystem I (PS I), PS II complex (PS II), cytochrome b6/f complex, photosynthetic electron transport and ATP synthase. The center of action for PS I is the pigment molecule P700, and the PS I subunits *PsaA* and *PsaB* are the key genes that regulate the synthesis of P700 chlorophyll a apolipoprotein A1 and P700 chlorophyll a apolipoprotein A2. Transcriptome analysis revealed that the expression of the *PsaA* and *PsaB* genes was up-regulated in diploids relative to triploids. PS II is a photosynthetic unit in the thylakoid membrane and contains two light-harvesting complexes and a photoreaction center. The light-harvesting complex that constitutes PS II is called light-harvesting complex II (LHCII), and the photoreaction center pigment of PS II is called P680. This is because the PS II reaction center pigment (P) absorbs light with a wavelength of 680 nm. Both the P680 response center D1 protein gene *PsbA* and the cytochrome b subunit *αPsbE* gene were up-regulated. Similarly, the CP43 chlorophyll apolipoprotein gene *PsbC* and oxygen evolution-enhancing protein gene *PsbO* also showed up-regulation. Cytochrome b6 generally exists as a dimer, and each monomer contains four different subunits. *PetB* is a key gene involved in the synthesis of the cytochrome b6f complex, and triploid *PetB* is up-regulated compared to the diploid. Ferredoxin (Fd) is widely present in various plants and participates in electron transfer. In the photoreaction step in photosynthesis, photosynthetic Fd receives electrons from PS I and transmits these electrons to various downstream metabolic processes dependent on Fd, which plays a key role in regulating the distribution of photosynthetic electrons to the metabolic pathways that depend on these electrons. The expression levels of photosynthetic electrons such as PetE, PetF, PetH and PetJ were all up-regulated. In ATP synthase, the F-type H+transport ATPase subunit β genes beta, alpha, delta, c, and b were also up-regulated, while gamma (2) and alpha (2) were down-regulated. The results of the study showed that the expression of most genes in the photosynthetic pathway of triploid tea plants

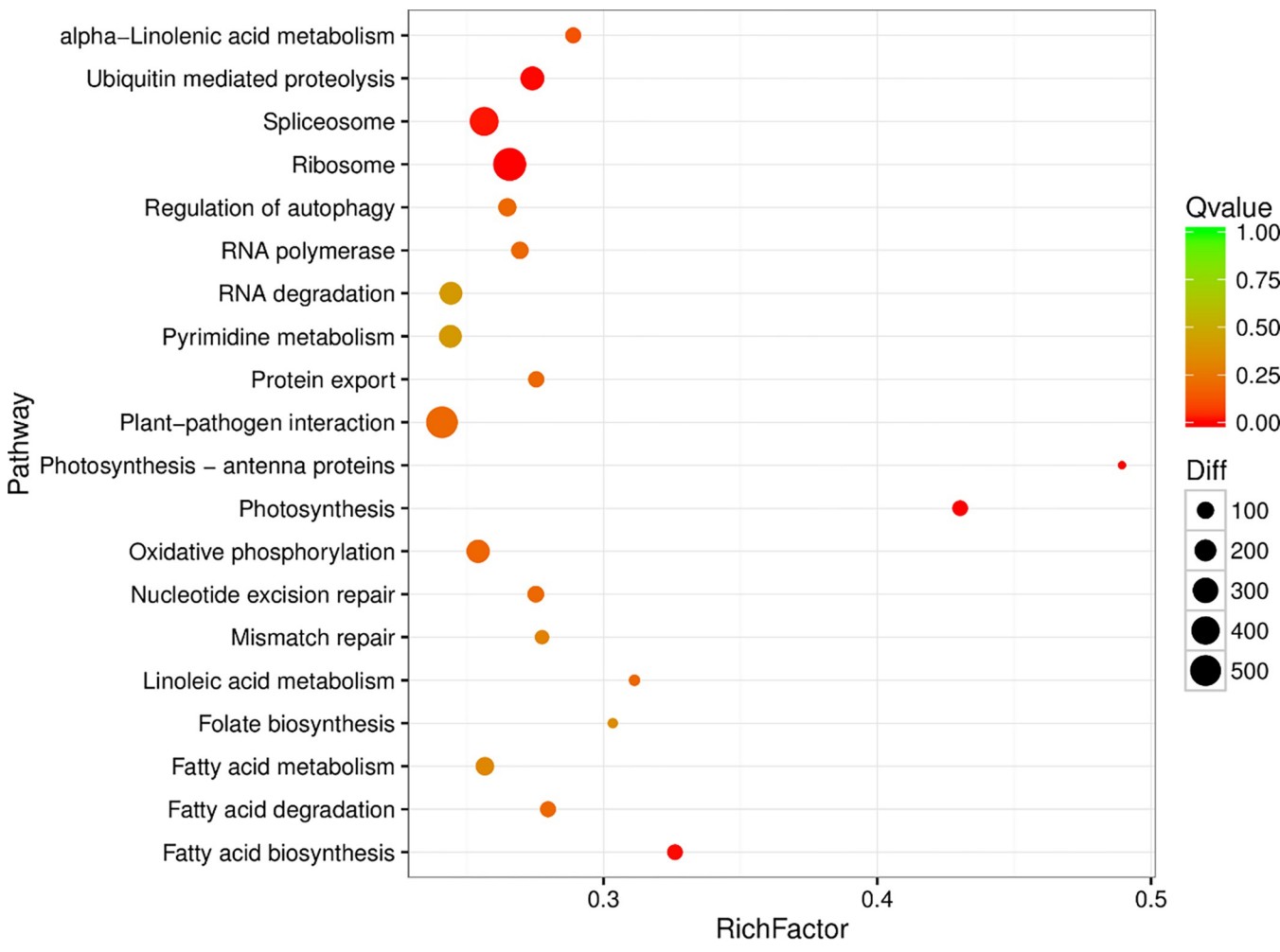

**Fig 8. Pathway enrichment analysis of differentially expressed genes.**

was up-regulated. These protein families were mapped to generalized photosynthetic pathways in the KEGG database (Fig 9). The transcripts covered the majority of the enzymes in the KEGG photosynthesis pathway.

## Analysis of key gene expression in stomatal development

In order to analyze the differences in stomatal development in the diploid and triploid leaves of tea plants, we identified 16 differentially expressed stomatal-related genes (P < 0.005). Differences in the expression of these genes led to changes in triploid stomatal density and size. Nine genes for negative-regulatory factors, which are key regulators of the stomata in plants, were identified. In the negative regulatory family, *SDD1*, *SERK1*, *2* and *EPF1*, *2* have a negative regulatory effect on stomatal development, while *EPLF9/ Stomagen* has a positive regulatory effect. The *SDD1* and *SERK1* genes were up-regulated and the *EPF1* gene was down-regulated (Table 2). *SERKs* can interact with the receptor-like protein Too Many Mouths (*TMM)* in a non-ligand-dependent manner to form a multiprotein receptor complex and negatively regulate stomatal development through signal transduction. These genes are involved in the biosynthesis and signal transduction of stomatal development by participating in different biological processes. For example, during stomatal development, cysteine-rich secretory

**Table 1. Differentially expressed genes involved in photosynthesis pathway.**

| GeneID | Enzyme | log2Fold | pvalue | KO_id | KO_Idefinition |
|---|---|---|---|---|---|
| CL16375.Contig1_All | Alpha | 5.59 | 1.06E-167 | K02111 | F-type H+-transporting ATPase subunit alpha |
| CL13329.Contig1_All | PsbS | 7.02 | 9.89E-52 | K02712 | photosystem II PsbK protein |
| CL1711.Contig6_All | PetB | 7.29 | 1.07E-27 | K02635 | cytochrome b6 |
| CL3166.Contig5_All | PsaA | 6.26 | 3.80E-27 | K02689 | photosystem I P700 chlorophyll a apoprotein A1 |
| CL1711.Contig8_All | PsbM | 5.90 | 2.08E-22 | K02704 | photosystem II CP47 chlorophyll apoprotein |
| CL3166.Contig1_All | PsaA | 3.49 | 2.95E-22 | K02689 | photosystem I P700 chlorophyll a apoprotein A1 |
| CL5827.Contig1_All | beta | 2.93 | 2.31E-21 | K02112 | F-type H+-transporting ATPase subunit beta [EC:3.6.3.14] |
| CL3044.Contig8_All | PsbC | 3.37 | 1.53E-19 | K02705 | photosystem II CP43 chlorophyll apoprotein |
| CL1711.Contig1_All | PsbB | 3.08 | 2.20E-18 | K02704 | photosystem II CP47 chlorophyll apoprotein |
| CL1711.Contig5_All | PsbE | 6.66 | 1.71E-14 | K02707 | photosystem II cytochrome b559 subunit alpha |
| Unigene33022_All | PsbR | 2.64 | 2.34E-14 | K03541 | photosystem II 10kDa protein |
| Unigene42313_All | PsaN | 2.89 | 2.77E-14 | K02701 | photosystem I subunit PsaN |
| CL3166.Contig6_All | PsaA | 2.07 | 4.85E-14 | K02689 | photosystem I P700 chlorophyll a apoprotein A1 |
| Unigene1782_All | PsbS | 2.03 | 8.86E-14 | K03542 | photosystem II 22kDa protein |
| CL15100.Contig1_All | gamma | -3.12 | 2.22E-12 | K02115 | F-type H+-transporting ATPase subunit gamma |
| CL15002.Contig1_All | PetF | 4.31 | 5.19E-12 | K02639 | ferredoxin |
| CL1711.Contig12_All | PsbM | 4.80 | 1.96E-11 | K02704 | photosystem II CP47 chlorophyll apoprotein |
| Unigene16570_All | b | 6.61 | 8.10E-11 | K02109 | F-type H+-transporting ATPase subunit b |
| CL6775.Contig2_All | PetF | 1.93 | 1.16E-10 | K02639 | ferredoxin |
| CL2036.Contig1_All | PsaD | 4.27 | 1.73E-10 | K02692 | photosystem I subunit II |
| CL17143.Contig2_All | PsaE | 2.29 | 3.32E-10 | K02693 | photosystem I subunit IV |
| Unigene1410_All | PsbW | 3.89 | 7.20E-10 | K02721 | photosystem II PsbW protein |
| Unigene2565_All | PsbW | 3.84 | 7.24E-10 | K02721 | photosystem II PsbW protein |
| CL15528.Contig1_All | PetF | 1.60 | 8.06E-10 | K02639 | ferredoxin |
| CL20325.Contig1_All | PetF | 2.00 | 6.96E-09 | K02639 | ferredoxin |
| CL3815.Contig1_All | PsaA | 5.02 | 9.67E-09 | K02689 | photosystem I P700 chlorophyll a apoprotein A1!K00430 |
| CL13069.Contig1_All | PsaL | 3.80 | 9.68E-09 | K02699 | photosystem I subunit XI |
| CL1711.Contig11_All | PsbL | -6.79 | 1.63E-08 | K02703 | photosystem II P680 reaction center D1 protein |
| Unigene12502_All | PetF | 3.81 | 2.58E-08 | K02639 | ferredoxin |
| CL20325.Contig3_All | PetF | 1.70 | 2.60E-08 | K02639 | ferredoxin |
| CL20679.Contig2_All | PsaH | 2.82 | 2.92E-08 | K02695 | photosystem I subunit VI |
| CL7015.Contig7_All | PsbP | 2.45 | 6.08E-08 | K02717 | photosystem II oxygen-evolving enhancer protein 2 |
| CL13329.Contig2_All | PsbK | 3.47 | 9.85E-08 | K02712 | photosystem II PsbK protein |
| Unigene23028_All | PsaG | 3.24 | 2.34E-07 | K08905 | photosystem I subunit V |
| Unigene28160_All | PsbR | 1.38 | 4.43E-07 | K03541 | photosystem II 10kDa protein |
| Unigene28155_All | PsaO | 3.24 | 6.32E-07 | K14332 | photosystem I subunit PsaO |
| CL17620.Contig3_All | PsbP | 1.57 | 6.83E-07 | K02717 | photosystem II oxygen-evolving enhancer protein 2 |
| CL7015.Contig4_All | PsbP | -6.19 | 7.81E-07 | K02717 | photosystem II oxygen-evolving enhancer protein 2 |
| CL4221.Contig1_All | PetF | 2.34 | 1.17E-06 | K02639 | ferredoxin |
| CL10041.Contig1_All | b | 3.23 | 1.52E-06 | K02109 | F-type H+-transporting ATPase subunit b |
| Unigene22289_All | PsbY | 2.98 | 1.62E-06 | K02723 | photosystem II PsbY protein |
| CL7061.Contig1_All | PsaK | 3.32 | 2.42E-06 | K02698 | photosystem I subunit X |
| CL5748.Contig2_All | b | 4.25 | 4.77E-06 | K02109 | F-type H+-transporting ATPase subunit b |
| CL21062.Contig2_All | PsbP | 2.51 | 4.85E-06 | K02717 | photosystem II oxygen-evolving enhancer protein 2 |
| CL6775.Contig1_All | PetF | 1.50 | 1.12E-05 | K02639 | ferredoxin |
| CL706.Contig8_All | PsbT | 6.12 | 1.19E-05 | K02714 | photosystem II PsbM protein |
| Unigene48598_All | PsbO | 1.34 | 1.19E-05 | K02716 | photosystem II oxygen-evolving enhancer protein 1 |

*(Continued)*

**Table 1.** (Continued)

| GeneID | Enzyme | log2Fold | pvalue | KO_id | KO_Idefinition |
|---|---|---|---|---|---|
| CL18974.Contig1_All | PetC | 2.84 | 1.44E-05 | K02636 | cytochrome b6-f complex iron-sulfur subunit |
| CL3421.Contig20_All | PetE | 2.72 | 2.02E-05 | K02638 | plastocyanin |
| Unigene27015_All | b | 2.43 | 2.15E-05 | K02109 | F-type H+-transporting ATPase subunit b |
| CL319.Contig1_All | a | -5.58 | 2.52E-05 | K02108 | F-type H+-transporting ATPase subunit a |
| CL15778.Contig1_All | Petc | 1.16 | 2.84E-05 | K02636 | cytochrome b6-f complex iron-sulfur subunit |
| CL21062.Contig1_All | PsbP | 1.95 | 3.21E-05 | K02717 | photosystem II oxygen-evolving enhancer protein 2 |
| CL7015.Contig6_All | PsbP | -3.18 | 3.57E-05 | K02717 | photosystem II oxygen-evolving enhancer protein 2 |
| CL20325.Contig2_All | PetF | 2.09 | 4.67E-05 | K02639 | ferredoxin |
| CL365.Contig9_All | Psb28 | 2.17 | 7.26E-05 | K08903 | photosystem II 13kDa protein |
| CL12923.Contig3_All | PsaA | 2.05 | 8.62E-05 | K02689 | photosystem I P700 chlorophyll a apoprotein A1 |
| Unigene27014_All | b | 3.92 | 0.0001508 | K02109 | F-type H+-transporting ATPase subunit b |
| CL20679.Contig1_All | PsaH | 1.96 | 0.0002048 | K02695 | photosystem I subunit VI |
| CL15528.Contig2_All | PetF | 1.03 | 0.0002294 | K02639 | ferredoxin |
| Unigene38304_All | PsbQ | 1.47 | 0.0002753 | K08901 | photosystem II oxygen-evolving enhancer protein 3 |
| Unigene7184_All | Psaf | 2.16 | 0.0002868 | K02694 | photosystem I subunit III |
| CL15100.Contig2_All | gamma | -1.58 | 0.0002878 | K02115 | F-type H+-transporting ATPase subunit gamma |
| CL6445.Contig1_All | PsaH | 2.67 | 0.0003723 | K02695 | photosystem I subunit VI |
| CL17620.Contig4_All | PsbP | 1.32 | 0.0004341 | K02717 | photosystem II oxygen-evolving enhancer protein 2 |
| CL1711.Contig26_All | PetB | 5.01 | 0.0005118 | K02635 | cytochrome b6 |
| CL16234.Contig1_All | PsbQ | 2.35 | 0.0005305 | K08901 | photosystem II oxygen-evolving enhancer protein 3 |
| CL3429.Contig2_All | PetH | 4.99 | 0.000553 | K02641 | ferredoxin—NADP+ reductase [EC:1.18.1.2] |
| Unigene10614_All | delta | 4.98 | 0.0005783 | K02113 | F-type H+-transporting ATPase subunit delta |
| Unigene46290_All | delta | 4.74 | 0.0011355 | K02113 | F-type H+-transporting ATPase subunit delta |
| CL365.Contig3_All | Psb28 | 1.54 | 0.0013349 | K08903 | photosystem II 13kDa protein |
| CL3429.Contig5_All | PetH | 4.67 | 0.0013843 | K02641 | ferredoxin—NADP+ reductase [EC:1.18.1.2] |
| CL2036.Contig2_All | PsaD | 2.14 | 0.0018953 | K02692 | photosystem I subunit II |
| Unigene12320_All | PsbL | 3.25 | 0.002214 | K02703 | photosystem II P680 reaction center D1 protein |

peptides belonging to the *EPF/EPFL* family act as ligands that interact with the corresponding receptors to transmit developmental signals, ensuring proper stomatal density and distribution. The key transcription factors involved in stomatal development in plants are basic-helix-loop-helix (*bHLH*)-type proteins, with *SPCH*, *FAMA* and *MUTE* playing important regulatory roles in stomatal development. There was no significant difference in expression between these three transcription factors in diploids and triploids. The *COP* and *HIC* genes are stomatal development factors that respond to light and $CO_2$ signaling; the *COP* gene was down-regulated and *HIC* was up-regulated.

## Identification and expression analysis of candidate genes involved in leaf development

The morphological structure of leaves appears simple, but the regulation mechanism for their development is very complicated. The final size of leaves is tightly controlled by environmental and genetic factors that must spatially and temporally coordinate cell expansion and cell cycle activity. In this study, we identified 28 putative genes associated with leaf development that belong to different pathways, including cell division, photosynthesis, transcription factors and auxin synthesis, which showed significant differential expression between diploids and triploids.

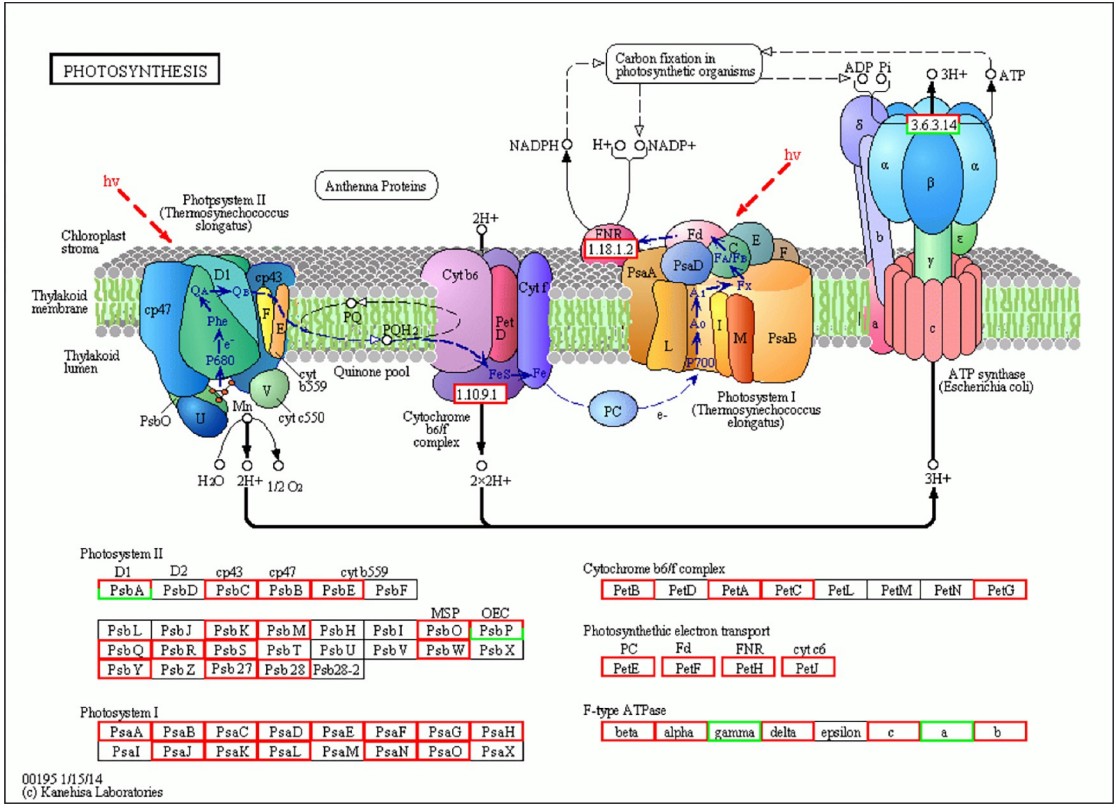

**Fig 9. Forty DEGs (red) involved in the photosynthesis pathway were up-regulated in QianMei 419 compared to QianFu 4.** DEGs (green) involved in the photosynthesis pathway were down-regulated in QianMei 419 compared to QianFu 4. Red and green boxes indicate genes whose expression levels were up and downregulated between QianMei 419 and QianFu 4. For interpretation of the references for the color in this figure legend, the reader is referred to the web version of this article.

Transcripts involved in the photosynthesis-photoreaction phase were observed to be differentially expressed, including genes encoding PS I, PS II, cytochrome b6/f complex, and ATP synthase (Fig 9). *PetB* is a key gene involved in the synthesis of the cytochrome b6f complex. *F1B* is an F-type H+ transport ATPase subunit β gene in ATP synthase. All these genes are functionally essential for carbon dioxide assimilation and were found to be up-regulated (Table 1) in the triploid leaves. To explore the intracellular transcriptional activity of diploid and triploid plants, we analyzed the expression of genes involved in the regulation of RNA polymerase and transcription initiation factors during transcription. The results showed that RNA polymerase I, RNA polymerase II and the transcription initiation factor were up-regulated in triploids. The *RPB2* gene, *TFIIA1* gene regulating RNA polymerase II and the transcription initiation factor were significantly up-regulated by about 2.5-fold in triploids compared to those in diploids (Table 3).

The expression of cell cycle-associated genes that regulate cell division can alter the organ volume of a plant, resulting in an increase in the number of cells and expansion of the cell volume in time and space. Cyclin-dependent kinases (*CDKs*), also known as cell cycle engine molecules, play a central role in the regulation of cell cycle function. The *CDK1* gene was up-regulated six-fold in triploid leaves compared to that in diploids (Table 3). The *CDK1* gene plays a key role in controlling the eukaryotic cell cycle by regulating centrosome circulation and mitosis initiation, promoting G2-M conversion, and regulating G1 and G1-S transformation by binding to multiple interphase cyclins. *CDK7* is a catalytic subunit of the *CDK-*

**Table 2. Differentially expressed genes (DEGs) related to stomatal development in diploid and triploid.**

| | Alias | Tea Genes | Putative function | Fold change | P-value |
|---|---|---|---|---|---|
| regulatory factors | SDD1 | Unigene48204_All | Stomatal density and distribution 1 | 6.2 | 0.0001 |
| | SDD1 | Unigene64413_All | Stomatal density and distribution 1 | 4.4 | 0.002 |
| | SDD1 | Unigene44141-All | Stomatal density and distribution 1 | 2.6 | 0.0007 |
| | EPFL9/ Stomagen | CL14931.Contig2_All | Epidermal patterning factors 9 | -1.6 | 0.001 |
| | EPFL9/ Stomagen | CL14931.Contig3_All | Epidermal patterning factors 9 | -1.7 | 0.0002 |
| | EPF1 | CL20193.Contig1_All | Epidermal patterning factors 1 | -3.4 | 0.0003 |
| | EPF2 | Unigene16900_All | Epidermal patterning factors 2 | -3.7 | 0.00001 |
| | SERK1 | Unigene17364_All | somatic embryogenesis receptor kinase family RLKs SERK) family RLKs | 2.3 | 0.00003 |
| | SERK2 | Unigene27737_All | somatic embryogenesis receptor kinase family RLKs | 3.3 | 0.0002 |
| signalling molecules | COP1 | CL9988.Contig3_All | Constitutive Photomorphogenesis 1 | -5.6 | 0.00007 |
| | COP10 | CL16283.Contig2_All | Constitutive Photomorphogenesis 10 | -2.6 | 0.00001 |
| | HIC | CL494.Contig10_All | High Carbon Dioxide | 8.1 | 0.00001 |
| bHLH transcription factor | MUTE | CL4078.Contig2_All | bHLH family transcription factor | 1.3 | 0.00001 |
| | FAMA | Unigene32561_All | bHLH family transcription factor | -0.7 | 0.009 |
| | SPCH | CL2555.Contig4_All | bHLH family transcription factor | -0.6 | 0.006 |
| receptor protein | TMM | Unigene32601_All | TOO MANY MOUTHS OS | -1.02 | 0.03 |

activated kinase (*CAK*) complex that regulates cell cycle progression. The relative expression of *CDK7* gene was up-regulated by 22.66% in triploid leaves of tea (Table 3). The serine/threonine kinase *BUB1* gene is involved in cell cycle control and RNA polymerase II-mediated RNA transcription, and was also up-regulated in triploids, with approximately twice the expression of diploid genes. Taken together, these results provide a framework for the regulatory network for leaf development in diploid and triploid leaves.

## Discussion

According to the photosynthetic characteristics of tea plants, we analyzed the photosynthetic pathways in the transcriptome data (Fig 6). From the photosynthetic pathways, we can see that PS I, PS II, cytochrome b6/f complex, photosynthetic electron transport and ATP synthase were involved in the up-regulation of key genes (*Alpha, PsbS, PetB, beta, PsbC, PsbB, PsbE, PsbR, PsaN, PsaA, PsbS, PsbM, b, PetF, PsaD, PsaE, PsbW, PsaL, PsbL, PetF, PsaH, PsbP, PsbK, PsaG, PsbR, PsaO, PsbY, PsaK, PsbT, PsbO, PetC, PetE, Psb28, PsbQ, Psdel*). The improvements in the photosynthesis of triploid tea plants, combined with the leaf functions, photosynthetic characteristics and key photosynthetic genes indicate that the photosynthetic capacity of the triploids was greater than that of diploids, which can increase yield.

**Table 3. Differentially expressed genes involved in leaf developmental growth in diploid and triploid.**

| | Alias | Tea Genes | Putative function | Fold change | P-value |
|---|---|---|---|---|---|
| Cell division | CDK1 | CL7926.Contig1_All | cyclin-dependent kinase 1 | 6.7 | 9.94E-07 |
| | CDK7 | CL19785.Contig2_All | cyclin-dependent kinase 7 | 1.2 | 9.94E-07 |
| | BUB1 | CL19622.Contig1_All | serine/threonine-protein kinase | 2.2 | 7.64E-12 |
| | CDC6 | CL9643.Contig2_All | cell division control protein 6 | 5.1 | 9.44E-05 |
| Regulation of transcription | TFIIA1 | CL11362.Contig6_All | transcription initiation factor | 2.6 | 6.38E-08 |
| | RPABC5 | Unigene4377_All | DNA-directed RNA polymerases I | 1.3 | 0.03 |
| | RPB2 | CL12803.Contig1_All | DNA-directed RNA polymerases | 2.6 | 0.01 |
| Plant hormone biosynthesis and signal transduction | AUX1 | CL6041.Contig1_All | 1.208807 | 2.66E-05 | |
| | ARF2 | CL1800.Contig7_All | auxin response factor 2 | 1.8 | 0.0001 |
| | ARF6 | CL2073.Contig8_All | auxin response factor 6 | 9.9 | 1.67E-21 |
| | ARF6 | Unigene643_All | auxin response factor 6 | 4.5 | 0.0001 |
| | ARF8 | CL1126.Contig2_All | auxin response factor 8 | 3.6 | 0.001 |
| | ARF19 | CL15534.Contig3_All | auxin response factor 19 | 2.0 | 7.32E-14 |
| | ARF19 | CL15534.Contig4_All | auxin response factor 19 | 3.2 | 6.14E-10 |
| | LAX1 | CL6041.Contig6_All | Like AUXIN RESISTANT1 | 1.5 | 0.007 |
| | LAX4 | Unigene18078_All | Like AUXIN RESISTANT4 | 3.2 | 0.003 |
| | BR60X1 | CL11134.Contig2_All | Brassinosteroid -6-oxidase | 1.4 | 0.001 |
| | BIG | CL11767.Contig5_All | Auxin transport protein | 2.3 | 9.17E-09 |
| | PIN3 | CL14227.Contig2_All | Auxin efflux carrier family protein | 1.45 | 0.007 |
| | BRI1 | CL5770.Contig2_All | BRassinosteroid Insensitive 1 | 1.962312 | 1.78E-07 |
| | BRI1 | CL2868.Contig3_All | BRassinosteroid Insensitive 1 | 3.274761 | 2.67E-06 |

Plant polyploidization allows cells to have additional genomic genetic material. The ratio of nuclear genetic material to cell size in eukaryotes, that is, the ratio of nucleus to cytoplasm, is usually fixed in eukaryotes [12,17]; thus, an increase in the genetic material in the nucleus often causes changes in cell size. Generally, as the ploidy of the chromosome increases, the volume of the plant cell and nucleus increases correspondingly. Plant tissues and organs also increase, along with the plant leaf growth morphology index, pollen size, seed size and stem diameter. This characteristic of polyploidy is called giantness. The immensity of polyploid organs leads to advantages in growth. The autotriploid *Populus tomentosa* fertilized by Mash-kina with artificially obtained unreduced pollen far exceeded the diploid of the same seedling age in terms of growth. The eight-year seedling height, diameter at breast height, and single plant volume of the eight-year seedling stage of the Chinese white poplar allogeneic triploid B301 under the same growth conditions as the Chinese white poplar diploid control were respectively higher by 39%, 72% and 255% [18]. The same experimental results were obtained in the current study. We found that triploid tea leaves have obvious growth advantages compared with diploid tea leaves, and the leaf length, leaf width and leaf area were increased by 23.37%, 41.12 and 59.81%, respectively, compared with diploid leaves, with very significant differences. The enlargement of plant organs generally occurs due to an increase in the number of cells or an increase in cell volume. Previous studies have found that the increase of plant cell ploidy will indeed promote the increase of cell volume [1,2].

We observed in the paraffin sections of diploid and triploid tea tree leaves that the xylem cells in the leaf veins of triploid tea tree leaves were larger than those of diploids, and the number of xylem cell layers was larger. The anatomical structure of the leaf mesophyll of diploid and triploid tea plants showed that the shape and arrangement of the triploid palisade tissue and sponge tissue cells in the same part of the leaves were significantly different, and the

intercellular space of the triploid sponge tissue had increased. Polyploidy increases the gene dose due to the doubling of chromosomes, and the amount of transcription and expression products will inevitably change accordingly. Plant hormones play an important role in the growth and development of plant organs. Analysis of tea plant gene expression revealed that the expression levels of brassinosteroid-6-oxidase (BR60X1) and its receptor protein (brassinosteroid insensitive 1, BRI1) genes were up-regulated and that the main physiological role of BRs is to promote cell elongation and division [19]. The length of leaves is regulated by BRs, which affect the polar expansion of cells [20,21], indicating that BRs may also affect the growth and development of the cell structure of triploid leaves. AVP1 can regulate the transport of auxin, and ARF6 can mediate the synthesis of auxin [22,23]. Overexpression of AVP1 and ARF6 genes can increase the number of cells, thereby increasing leaf size [24,25]. The regulation of plant leaf growth and development is jointly affected by plant hormones and cell division.

Transcription factors are regulatory molecules for gene expression that bind to either the promoter or enhancer regions of a gene and up- or downregulate its expression [26,27]. They form a complex system that controls cellular growth, differentiation, genetic responses to the environment, and organismal development and evolution [28]. Based on our results, *GIF*, *GRF* and *TCP* transcription factors showed significant up-/downregulation in triploids, indicating that these transcription factors are involved in the growth and development of triploid leaves (Table 4). Studies have shown that overexpression of *GIF* in plants results in leaves that are larger and contain more cells [25,29]. GIF1 protein interacts with the growth regulators GRF1, GRF2 and GRF5 to regulate cell division, leaf growth and development. The *GIF1* protein is involved in the regulation of leaf growth by interacting with members of the putative GIF transcription factor family, *GRF1*, *GRF2* and *GRF5*, to regulate cell proliferation [30]. Horiguchi et al. found that overexpression of *GRF5* increased the cell number, resulting in increased leaf area [25]. In these plants, the initial size and growth of the leaves did not change, but growth at later stages was faster, and the duration of growth was prolonged, indicating that the *GRF5* gene is mainly involved in the late stage of leaf growth and development [31,32]. As transcriptional coactivators of *GRF* proteins, *GIF* proteins also contribute to leaf development. In *Arabidopsis* leaf development, GIF is essential for cell division [31,33]. The Teosinte branched 1/cycloidea/proliferating cell factor (TCP) family is a family of transcription factors associated with leaf development. Studies have shown that the *Arabidopsis thaliana TCP* gene plays a role in leaf cell growth and division. *AtTCP20* is involved in cell division, cell expansion and growth coordination [24,34]. In the *Arabidopsis* jaw-D mutant, miR319 is overexpressed, inhibiting the expression of *TCP2*, *TCP3*, *TCP4*, *TCP10* and *TCP24*, and forming large and crinkled leaves [35]. *AtTCP* seems to have antagonistic effects on cell growth and division [36]. Ectopic expression of *AtTCP3* inhibits the formation of shoot tip meristems [37]. The *TCP* transcription factor family *TCP2*, *TCP3*, *TCP4*, *TCP14*, *TCP15*, *TCP20* and *TCP24* genes were downregulated in tea triploids, indicating that the down-regulation of the *TCP* gene promoted the division and growth of triploid leaf cells (Table 4). Besides transcription factors, other regulators can affect the duration of cell division during leaf development. Plant hormones play an important role in the growth and development of plant organs [32,35]. The auxin-regulated involved in organ size (*ARGOS*) gene is induced by auxin expression and plays a role in the positive regulation of cell division and the expansion of the leaf. Overexpression of the *ARGOS* gene in *Arabidopsis* promotes plant leaf enlargement, while loss of *ARGOS* gene function causes the leaf size to decrease. In maize, overexpression of the zea mays *ARGOS1* (*ZAR1*) enhances the leaf, stalk and ear sizes as well as grain yield due to increased cell number and promotes drought-stress tolerance [38–40].

**Table 4. Transcription factor identified in the DEGs involved in leaf developmental in diploid and triploid.**

| | Alias | Tea Genes | Putative function | Fold change | P-value |
|---|---|---|---|---|---|
| Transcription factor | TCP2 | Unigene6055_All | TEOSINTE BRANCHED1/CYCLOIDEA/PCF | 4.5 | 0.002 |
| | TCP3 | Unigene43100_All | TEOSINTE BRANCHED1/CYCLOIDEA/PCF | 2.83923 | 0.001 |
| | TCP4 | Unigene26990_All | TEOSINTE BRANCHED1/CYCLOIDEA/PCF | 1.17505 | 0.08 |
| | TCP7 | Unigene28095_All | TEOSINTE BRANCHED1/CYCLOIDEA/PCF | 2.46923 | 1.02E-05 |
| | TCP11 | Unigene16745_All | TEOSINTE BRANCHED1/CYCLOIDEA/PCF | 2.85517 | 2.08E-06 |
| | TCP14 | Unigene11723_All | TEOSINTE BRANCHED1/CYCLOIDEA/PCF | 2.724 | 4.93E-05 |
| | TCP15 | CL1762.Contig2_All | TEOSINTE BRANCHED1/CYCLOIDEA/PCF | 4.24584 | 0.004 |
| | TCP15 | Unigene11723_All | TEOSINTE BRANCHED1/CYCLOIDEA/PCF | 2.724 | 4.93E-05 |
| | TCP20 | Unigene45934_All | TEOSINTE BRANCHED1/CYCLOIDEA/PCF | 3.02894 | 0.05 |
| | TCP20 | CL1598.Contig1_All | TEOSINTE BRANCHED1/CYCLOIDEA/PCF | 3.49031 | 0.02 |
| | TCP24 | Unigene6055_All | TEOSINTE BRANCHED1/CYCLOIDEA/PCF | 4.45414 | 0.002 |
| | GRF1 | CL766.Contig8_All | GROWTH-REGULATING FACTOR | -3.976247 | 2.05E-06 |
| | GRF5 | CL6110.Contig2_All | GROWTH-REGULATING FACTOR | -2.261885 | 0.0144514 |
| | GRF7 | CL13904.Contig1_All | GROWTH-REGULATING FACTOR | -3.477404 | 0.000316 |
| | GRF9 | CL13904.Contig2_All | GROWTH-REGULATING FACTOR | -3.344828 | 0.002714 |
| | GIF1 | CL766.Contig8_All | GRF1-interacting factor 1 | -3.976247 | 2.05E-06 |
| | ARGOS | Unigene22883_All | AUXIN-REGULATED GENE INVOLVED IN ORGAN SIZE | -2.03898 | 3.67E-05 |

Genes that regulate the cell cycle can affect the morphology and size of plant organs. CDKs are called cell cycle engine molecules and play a core regulatory role in the operation of the cell cycle. Overexpression of KRP1 and KRP2 can inhibit the activity of CDKs, such that *Arabidopsis* cell division is slowed, the number of cells is reduced, and the organ volume of the plant becomes smaller. Transcriptome results showed that the CDK1 gene was up-regulated more than six-fold in triploid tea trees compared with the gene in diploids (Table 2). The CDK1 gene plays a key role in controlling the eukaryotic cell cycle by regulating centrosome circulation and the initiation of mitosis. It also promotes G2-M transition and regulates the G1 and G1-S transitions by binding to multiple interphase cyclins. CDK7 is the catalytic subunit of the CAK complex, which can regulate the cell cycle process. The expression of the CDK7 gene in triploid tea leaves was up-regulated by 22.66% relative to the expression in diploid leaves (Table 3). The serine/threonine kinase BUB1 gene is involved in cell cycle control and RNA polymerase II-mediated RNA transcription, and was up-regulated in triploids by about double the expression in diploids. Cdc6 is a key regulator of DNA replication in eukaryotes. The expression level of Cdc6 in the leaves of triploid tea plants was five times greater than that of

diploids (Table 3). Therefore, the cell division activity of triploid leaves was stronger than that of diploid leaves. The above analysis shows that plant hormones and cell division-related genes jointly regulate the growth and development of tea leaves.

The stomata in the epidermal cells of plant leaves are the main channels for $CO_2$ and $H_2O$ to enter and exit the leaves. Stomata are generally composed of two guard cells and an intermediate pore. The pore size and the degree of opening and closing directly determine the Tr and photosynthesis of the plant. Jia Ti's research demonstrated that an increase in the thickness of the palisade tissue of apple leaves has an important effect on the photosynthesis rate of its leaves and the increase of its yield [10]. In addition, the palisade tissue of Pingou hybrid hazel leaves can effectively reduce water Tr in the leaves and improve the photosynthetic efficiency of the plant [12]. The thicker the palisade tissue of the leaf is, the stronger the water retention capacity it exhibits. We speculated that the stomatal development state may affect the photosynthetic rate to a certain extent. In this study, we found that the stomata size of the triploid was significantly larger than that of the diploid, and the leaf mesophyll palisade tissue cells in triploids were arranged more closely than those of diploid tea leaves, with small intercellular spaces (Fig 1). The spongy tissue cells were loosely arranged and the intercellular spaces were larger. Chloroplast carbon dioxide gas enters from the stomata through the intercellular spaces to reach the surface of mesophyll cells, which are part of the internal area of leaves and are covered by chloroplasts. The size of the cell surface area exposed to the intercellular space is related to photosynthetic functiont. The development of the stomata is accompanied by leaf growth. Studying the relationship between stomata morphology and photosynthetic function during the development of plant leaves is conducive to a comprehensive understanding of the entire life process of plant leaves. In studying the diploid and triploid tea leaf stomata, we found that the stomata density of diploid leaves was significantly higher than that of triploids, and the stomata size of triploids was significantly larger than that of diploids (Fig 3). An increase in the number of stomata and the size of the stomata in each leaf has an important effect on gas exchange in the leaves, and a decrease in stomata density is mainly attributed to the growth of the leaf area. The enlarged stomata of the triploid indicated that the leaves could absorb $CO_2$ more effectively and the carbon sequestration ability was stronger. Analysis of the differences in key genes at each stage of photosynthesis in diploid and triploid tea leaves revealed that photosynthesis in triploid tea leaves involved PS I, PS II, cytochrome b6/f complex and ATP synthesis. Most of the key genes in enzymes were up-regulated (Fig 9), consistent with the observation of increased leaf fence tissue thickness, which played an important role in the increase of the leaf photosynthetic rate and yield. These findings indicated that the triploid tea plant leaves had enlarged stomata, tightly arranged palisade tissues, loosely arranged sponge tissue cells, and large intercellular spaces, which is conducive to the absorption and utilization of $CO_2$ by photosynthesis.

Plant stomata are small holes surrounded by a pair of guard cells. Stomatal development is generally regulated by a three-step transcriptional cascade of three structurally similar bHLH transcription factors: *SPCH*, *MUTE* and *FATA* [41]. Analysis of the diploid and triploid transcription results in tea showed no difference in *SPCH* and *FAMA* regulatory factor gene expression, although *MUTE* regulatory factors were up-regulated, but not significantly so. This indicated that the differences in triploid stomatal development were not related to the three regulatory factors.

There are many negative regulatory factors involved in stomatal development in plants [42], such as the epidermal model factor *EPF*, the leucine-rich receptor-like protein *TMM*, the subtilisin-like *SDD1* and the ERECTA family (*ERf*) of receptor-like kinases. A family of secreted peptide signals known as Epidermal Patterning Factors (or *EPFs*) has been proposed to compete for putative cell surface receptors, believed to comprise *TMM* and a putative

leucine-rich repeat receptor-like protein kinase [43]. *EPF1* [44] and *EPF2* [45,46] have been identified as negative regulatory factors in stomatal development. Evidence suggests that receptor binding activates an intracellular mitogen-activated protein kinase cascade that phosphorylates and destabilizes a *bHLH* transcription factor required early in leaf development for cells to enter the stomatal lineage [47]. *EPF2* and *EPF1* have very similar amino acid sequences and can inhibit the development of stomata by blocking the production of meristemoid cells. *EPF2* regulates the differentiation of the protodermal cell to meristemoid mother cells, whereas *EPF1* regulates the direction of the spacing division that generates satellite meristemoids. When *EPF2* is overexpressed, plant epidermis cannot form stomata, and stomatal and non-stomatal epidermis cells aggregate easily [42]. This may indicate that the gene can block the development of the initial stomatal system. The Sugano [48] group studied the regulatory factors favorable for stomatal development in *Arabidopsis thaliana* and found that *Stomagen* and *EPF1/EPF2* competed with each other and that *Stomagen* was able to bind to the *TMM* receptor protein, which is part of the negative regulatory family but has a positive effect on stomatal development. Overexpression of the *Stomagen* gene increases the number of stomata. Conversely, inhibition of the *Stomagen* gene reduces the number of stomata [49]. *SERKs* can interact with *TMM* in a non-ligand-dependent manner to form a multiprotein receptor complex and negatively regulate stomatal development through signal transduction.

In the negative regulatory family, *SDD1* is negatively regulated by stomatal development independent of other signaling pathways [50]. The *SDD1* mutant increases stomatal density and forms stomatal clusters [49]. *SDD1* is believed to proteolytically process certain negative signaling factors, such as *EPF1* and *EPF2*. However, overexpression of either *EPF1* or *EPF2* in the SDD1 background reduced stomatal density to the same level as in wild-type plants, suggesting that the function of these signaling peptides is independent of *SDD1*. It is possible that negative signaling receptors (*TMM* and *ERf*) are modulated by *SDD1* [42,44,45]. The *SDD1* gene was significantly up-regulated in triploid tea leaves (Table 2). This indicates that SDD1 negatively regulates the stomatal development of the leaves of triploid tea, which reduces the stomatal density of the leaves of triploids (Fig 2).

The regulation of stomatal development is also influenced by environmental factors as a whole. COP1 and HIC play important roles in the regulation of stomatal development by light and $CO_2$, respectively [51]. As an E3 ubiquitin ligase, COP1 can inhibit the expression of through photoreceptors, and COP1 deletion mutants can cause stomata to form clusters. The COP1 gene in triploid tea leaves was up-regulated by about five-fold compared with that in diploid tea leaves; hence, the size of the stomata in the triploid tea leaves was larger, and the decrease in number was consistent with the results of the study. The HIC-encoded 3-oxidized phthalyl-CoA synthase gene was significantly down-regulated in triploid leaves [52–54]. HIC is involved in the formation of epidermal waxy aggregates. The HIC gene in plants is very sensitive to $CO_2$ concentration. If the $CO_2$ concentration is abnormally high, the gene will be down-regulated to inhibit the formation of stomata. In summary, plant stomatal development is closely regulated at multiple levels by a large number of genes as well as self-development programs and environmental signals.

## Conclusions

In this study, we examined the functional traits and photosynthetic characteristics of diploid and triploid tea leaves, and investigated the molecular mechanisms behind these differences through transcriptome analysis. Potential links between gene expression and cytological changes in tea leaves were observed in diploid and triploid tea leaves. Triploid tea had larger stomata than diploid tea. From the analysis of the leaf microstructure, we found that the xylem

cells in the veins of the triploid tea tree leaves were larger than those in the diploid trees, the number of xylem cells was greater and the sponge tissue gap was larger. Comparative transcriptome analysis shows that compared to diploid trees, the expression of genes involved in photosynthesis and cell division was higher in triploids, especially the key enzymes and transcription factors that work at the branch point of the cell formation pathway, which may explain the differences in morphological characteristics and photosynthetic efficiency. The transcriptome data and leaf microstructure obtained in this study will enhance our understanding of the molecular mechanism of tea and stomata development and provide a basis for molecular breeding of high-quality and high-yield tea varieties.

## Supporting information

**S1 Table. Primers used in quantitative real-time PCR.**
(XLS)

**S2 Table. DEG KEGG pathway.**
(XLS)

**S3 Table. Number of up-regulated genes and down-regulated genes in differential genes.**
(PATHWAY)

## Acknowledgments

We thank Professor Lu Litang of the Tea School of Guizhou University for revising the manuscript.

## Author Contributions

**Data curation:** Xinzhuan Yao.

**Formal analysis:** Litang Lu.

**Funding acquisition:** Litang Lu.

**Methodology:** Xinzhuan Yao.

**Software:** Hufang Chen, Baohui Zhang.

**Supervision:** Zhengwu Chen.

**Writing – original draft:** Xinzhuan Yao, Yong Qi.

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
