## [Decision Letter · Decision Letter 0]

3 Aug 2022

PONE-D-22-18975Study of Camellia sinensis diploid and triploid leaf development mechanism based on transcriptome and leaf characteristicsPLOS ONE

Dear Dr. Xinzhuan yao,

Thank you for submitting your manuscript to PLOS ONE. After careful consideration, we feel that it has merit but needs minor revision. Therefore, we invite you to submit a revised version of the manuscript that addresses the points raised during the review process.

Please submit your revised manuscript in 10 days. If you will need more time than this to complete your revisions, please reply to this message or contact the journal office at plosone@plos.org. Please include the following items when submitting your revised manuscript:A rebuttal letter that responds to each point raised by the academic editor and reviewer(s). You should upload this letter as a separate file labeled 'Response to Reviewers'.A marked-up copy of your manuscript that highlights changes made to the original version. You should upload this as a separate file labeled 'Revised Manuscript with Track Changes'.An unmarked version of your revised paper without tracked changes. You should upload this as a separate file labeled 'Manuscript'.If applicable, we recommend that you deposit your laboratory protocols in protocols.io to enhance the reproducibility of your results. Protocols.io assigns your protocol its own identifier (DOI) so that it can be cited independently in the future. For instructions see: https://journals.plos.org/plosone/s/submission-guidelines#loc-laboratory-protocols. Additionally, PLOS ONE offers an option for publishing peer-reviewed Lab Protocol articles, which describe protocols hosted on protocols.io. Read more information on sharing protocols at https://plos.org/protocols?utm_medium=editorial-email&utm_source=authorletters&utm_campaign=protocols.

We look forward to receiving your revised manuscript.

Kind regards,

Huseyin Tombuloglu

Academic Editor

PLOS ONE

Journal Requirements:

2. We noticed your current submission has significant text overlap with the following preprint:

https://www.researchsquare.com/article/rs-20102/v2

Please specify the relation and difference between the two studies. If they are the same study, please specify the reasons for the authorship change.

"No"

Additional Editor Comments:

Dear Dr. Xinzhuan yao,

On behalf of the reviewer`s comments, it is my pleasure to inform you that your manuscript "Study of Camellia sinensis diploid and triploid leaf development mechanism based on transcriptome and leaf characteristics" has been accepted for publication. However, before the final approval, I suggest to consider the minor comments of reviewer 2.

Dr. Huseyin Tombuloglu

Reviewers' comments:

Reviewer's Responses to Questions

**Comments to the Author**

1. Is the manuscript technically sound, and do the data support the conclusions?

Reviewer #1: Yes

Reviewer #2: Yes

2. Has the statistical analysis been performed appropriately and rigorously? 

Reviewer #1: Yes

Reviewer #2: Yes

3. Have the authors made all data underlying the findings in their manuscript fully available?

Reviewer #1: Yes

Reviewer #2: Yes

4. Is the manuscript presented in an intelligible fashion and written in standard English?

Reviewer #1: Yes

Reviewer #2: Yes

5. Review Comments to the Author

Reviewer #1: The authors have analyzed the leaf functional traits, photosynthetic characteristics, leaf cell structure and transcriptome of Camellia sinensis. They claim that there was a correlation in between the nuclei content and tea content. The article is interesting, well written, it brings new information since the study is carried out on the tea nuclei content not much studied

Reviewer #2: This manuscript studied functional traits and photosynthetic characteristics of diploid and triploid tea leaves, and investigated the molecular mechanisms through transcriptome analysis.

The subject of this study is my interest. The study is well documented and designed. Findings and data are important.

There are only minor points to review.

Results

Line 199: It was mentioned that ′the stomata density of triploid leaves was significantly lower than that of diploids; the stomata density of diploid tea leaves was twice that of triploid stomata′. But it was not explained well in discussion why the stomatal density is higher in diploids. It must be clarified.

Figures

The resolution of Figure 4 and 6 are not good and legible.

6. PLOS authors have the option to publish the peer review history of their article (what does this mean?). If published, this will include your full peer review and any attached files.

Reviewer #1: No

Reviewer #2: **Yes: **Guzin Tombuloglu

---

## [Author Response · Author response to Decision Letter 0]

12 Sep 2022

Journal Requirements:

Response:The manuscript has been revised in the style of PLOSE ONE

2. We noticed your current submission has significant text overlap with the following preprint:

https://www.researchsquare.com/article/rs-20102/v2

Please specify the relation and difference between the two studies. If they are the same study, please specify the reasons for the authorship change.

Response:The two studies are basically the same. The main reasons: 1) When submitting the BMC Plant biology journal, I accidentally selected the preprint version, and the manuscript was rejected, and the preprint version could not be cancelled. 2) We have made a lot of revisions and improvements to the manuscript, mainly completed by Yao Xinzhuan, Chen Hufang and Zhang Baohui, and the first author Qi Yong did not participate in the revision and improvement, so the order of authors changed.

Response:Thank you very much for your comments，Submission system modification.

"No"

Response: As required by the journal, it has been added to the recommendation letter.

Response: Response: As required by the journal, it has been added to the recommendation letter.

6.PLOS requires an ORCID iD for the corresponding author in Editorial Manager on papers submitted after December 6th, 2016. Please ensure that you have an ORCID iD and that it is validated in Editorial Manager. To do this, go to ‘Update my Information’ (in the upper left-hand corner of the main menu), and click on the Fetch/Validate link next to the ORCID field. This will take you to the ORCID site and allow you to create a new iD or authenticate a pre-existing iD in Editorial Manager. Please see the following video for instructions on linking an ORCID iD to your Editorial Manager account: https://www.youtube.com/watch?v=_xcclfuvtxQ

Response: As required by the journal, it has been added to the recommendation letter.

Response: Revised in the manuscript as required by the Journal

Comments to the author (if any): 

Reviewer #1: General comment 

This manuscript studied functional traits and photosynthetic characteristics of diploid and 

triploid tea leaves, and investigated the molecular mechanisms through transcriptome 

analysis. 

The subject of this study is my interest. The study is well documented and designed. Findings 

and data are important. 

There are only minor points to review. 

Results 

Line 199: It was mentioned that ′the stomata density of triploid leaves was significantly 

lower than that of diploids; the stomata density of diploid tea leaves was twice that of 

triploid stomata′. But it was not explained well in discussion why the stomatal density is 

higher in diploids. It must be clarified. 

Response:Thank you very much for your comments. In our manuscript, line 622-733, both explained that the stomatal density of triploid leaves was significantly.

Figures 

The resolution of Figure 4 and 6 are not good and legible. 

Response:Thank you very much for your comments. We have revised it in the manuscript.

---

## [Editor Report · Decision Letter 1]

21 Sep 2022

Study of Camellia sinensis diploid and triploid leaf development mechanism based on transcriptome and leaf characteristics

PONE-D-22-18975R1

Dear Dr. Lu,

We’re pleased to inform you that your manuscript has been judged scientifically suitable for publication and will be formally accepted for publication once it meets all outstanding technical requirements.

Kind regards,

Huseyin Tombuloglu

Academic Editor

PLOS ONE
---

## [Editor Report · Acceptance letter]

9 Feb 2023

PONE-D-22-18975R1 

Study of *Camellia sinensis* diploid and triploid leaf development mechanism based on transcriptome and leaf characteristics 

Dear Dr. Yao:

I'm pleased to inform you that your manuscript has been deemed suitable for publication in PLOS ONE. Congratulations! Your manuscript is now with our production department. 

Kind regards, 

on behalf of

Dr. Huseyin Tombuloglu 

Academic Editor

PLOS ONE